# Idiopathic Pulmonary Fibrosis and Lung Transplantation: When it is Feasible

**DOI:** 10.3390/medicina55100702

**Published:** 2019-10-19

**Authors:** Elisabetta Balestro, Elisabetta Cocconcelli, Mariaenrica Tinè, Davide Biondini, Eleonora Faccioli, Marina Saetta, Federico Rea

**Affiliations:** Department of Cardiac, Thoracic, Vascular Sciences and Public Health, University of Padova, Padova City Hospital, 35128 Padova, Italy; ecocconcelli@icloud.com (E.C.); mariaenrica.tine@gmail.com (M.T.); dav.biondini@gmail.com (D.B.); faccioli.eleonora@gmail.com (E.F.); marina.saetta@unipd.it (M.S.); federico.rea@unipd.it (F.R.)

**Keywords:** idiopathic pulmonary fibrosis, lung transplantation, acute exacerbation, comorbidities

## Abstract

Despite the availability of antifibrotic therapies, many patients with idiopathic pulmonary fibrosis (IPF) will progress to advanced disease and require lung transplantation. International guidelines for transplant referral and listing of patients with interstitial lung disease are not specific to those with IPF and were published before the widespread use of antifibrotic therapy. In this review, we discussed difficulties in decision-making when dealing with patients with IPF due to the wide variability in clinical course and life expectancy, as well as the acute deterioration associated with exacerbations. Indeed, the ideal timing for referral and listing for lung transplant remains challenging, and the acute deterioration might be influenced after transplant outcomes. Of note, patients with IPF are frequently affected by multimorbidity, thus a screening program for occurring conditions, such as coronary artery disease and pulmonary hypertension, before lung transplant listing is crucial to candidate selection, risk stratification, and optimal outcomes. Among several comorbidities, it is of extreme importance to highlight that the prevalence of lung cancer is increased amongst patients affected by IPF; therefore, candidates’ surveillance is critical to avoid organ allocation to unsuitable patients. For all these reasons, early referral and close longitudinal follow-up for potential lung transplant candidates are widely encouraged.

## 1. Introduction

Idiopathic pulmonary fibrosis (IPF) is a devastating form of chronic, progressive, fibrosing interstitial pneumonia of unknown cause, limited to the lung, which carries a poor prognosis with a median survival of 3–5 years from diagnosis [1,2,3] and a 5-year mortality rate of 60–80%. The disease course is highly heterogeneous and unpredictable and is characterized by progressive breathlessness and lung function decline. While most individuals with IPF experience a slow but steady worsening of their disease (slow progressors), others experience an accelerated progression (rapid progressors) [2,3,4]. Moreover, a minority of the patients may present an acute clinical worsening, named acute exacerbation [2,3,4]. In these patients, the mortality reaches up to 85%, and no treatment has been proven effective with randomized controlled trials; an uncontrolled cohort showed conflicting results with corticosteroids or immunomodulatory therapies [5].

Several predictors of disease progression have been studied, but only a few have been validated and routinely used in clinical practice. In particular, forced vital capacity (FVC) is a reproducible and easy-to-perform functional measure, and a predicted annual decline of >10% is associated with higher 5-year mortality [6]. Moreover, it has been frequently adopted as an end-point in clinical trials [7,8]. Another predictor of outcome is the fibrosis score at High-Resolution Computed Tomography (HRCT), which can be measured either visually or automatically [9,10,11]. In order to associate disease severity and outcome, composite scores that combine information from different domains (e.g., demographic, physiologic, radiologic, etc.) have been created. In particular, the gender-age-physiology (GAP) score [12], which incorporates gender (G) and age (A) to functional parameters (P), identifies three stages of increasing mortality risk. This score was later improved and became more accurate by the integration of the longitudinal change in lung function values rather than the single value (longitudinal GAP) [13], and by the replacement of functional parameters with the computed tomography (CT) fibrosis extent (CT GAP score) [14].

In late 2014, two therapies were approved for use in IPF: nintedanib and pirfenidone. These new agents provide notable breakthroughs in the management of IPF [15,16,17]. Both pirfenidone and nintedanib slow the lung function decline over time [17]. Therefore, it is important to inform patients that the ultimate goal of treatment is stabilization in terms of slower functional deterioration. Treatment effects are difficult to evaluate. If the patient is considered to be stable after 6 months of treatment, the drug may be judged effective, and the clinician should continue to deliver the therapy. However, treatment may be continued regardless of the results of the lung function tests at follow-up; in fact, in IPF, the lung function decline during a given period does not necessarily predict a decline in the following period [18,19]. If the rate of deterioration is unchanged or increased, referral to a lung transplantation center is recommended [17].

In this review, we dealt with specific challenges associated with lung transplantation, such as referral and listing criteria for transplant with a particular focus on acute exacerbation; then, we considered strategies for ensuring that suitable IPF patients are timely referred for lung transplantation; finally, we addressed the most relevant comorbidities that may impact on post-transplant outcomes.

## 2. Referral and Listing Criteria for Lung Transplant

Lung transplantation (LTx) remains a life-saving and life-prolonging procedure for end-stage IPF. Despite the availability of effective pharmacologic treatments for IPF [15,16], LTx is so far the only treatment with proven benefit on survival for carefully selected patients with advanced disease and respiratory failure. Due to the wide variability in clinical course and life expectancy [20,21], as well as the rapid and often catastrophic deterioration associated with acute exacerbations, the ideal timing for referral and listing for LTx remains challenging. Therefore, early referral and close longitudinal follow-up for potential candidates is strongly recommended. Patients with IPF should undergo LTx when their post-transplant life expectancy exceeds their current life expectancy without the transplant. Several indices, including lung function, functional status, and radiological severity, have been associated with poor prognosis and should be serially evaluated to capture the appropriate timing for referring patients to a transplant center. Given the difficulty in predicting which IPF patients have the highest mortality, the International Society for Heart and Lung Transplantation (ISHLT) proposed and recently updated referral and listing guidelines based on described prognostic indicators in IPF (Table 1) [22]. In addition, major and relative contraindications to transplantation are also explored in this report (Table 2). Of note, age major of 65 years is included in the relative contraindication to transplant because increasing age is generally associated with comorbid conditions that are either absolute or relative contraindications. IPF is defined as a disease of ‘aging’, and an increased number of organs are frequently allocated to patients older than 65 years, according to the improvement in lung transplant center expertise. Therefore, the upper limit for LTx is moving toward 70 years old and, even though long-term data on this patient are not available, the international consensus suggests that the “physiological age” counts more than a rigid age limit, meaning that a scrupulous screening of comorbidities, rehabilitation reserve, and social/family support would be the prior decision criteria.

Stable patients with bad prognostic factors should be promptly assessed for transplant suitability. Nevertheless, it should be considered that no randomized trial exists to validate these statements. Thus, the scientific community of each country has variably accepted these recommendations, and great discrepancies can be found when these expert opinions are translated into clinical practice. In order to standardize the candidate selection and assist the clinicians in the enlisting process, for instance, an Italian group has provided a national consensus statement on lung transplant patient selection [23]. Of interest, the Italian Ministry of Health attested 143 lung transplants in Italy in 2018, similar to the previous two years and more than in the previous years, with a mean waiting list time of 12 months (two months less than what registered in 2002). Ten transplant centers have been recognized in Italy, with Padua, Turin, and Milan—the three leader centers.

Nonetheless, given the debatable nature of these tools, careful clinical judgment by experienced health care professionals is important in taking care of this complex patient population. A simplified representation of all the steps, from diagnosis to transplant, throughout the heterogeneous course of the disease is proposed in Figure 1.

## 3. A Focus on Acute Exacerbation: A Good or a Bad Indication?

A recent international working group reported the definition and diagnostic criteria of acute exacerbation of idiopathic pulmonary fibrosis (AE-IPF) and defined it as an acute, clinically significant respiratory deterioration characterized by evidence of new widespread alveolar abnormality. The revised diagnostic criteria include a previous or concurrent diagnosis of IPF; an acute worsening or development of dyspnea typically less than 1 month duration; computed tomography with new bilateral ground-glass opacity and/or consolidation superimposed on a background pattern consistent with usual interstitial pneumonia pattern and a deterioration not fully explained by cardiac failure or fluid overload [5]. A systematic review of studies, reporting 1- and 3-month survival rates after AE-IPF, demonstrated a pooled mortality rate over eight studies of 60% and 67%, respectively [24].

Given the lack of successful pharmacological therapies for limiting the devasting outcome of this acute deterioration and the shortage of evidence-based effective respiratory assistance techniques, lung transplantation is still considered a potentially lifesaving therapy for those who develop an acute exacerbation. Indeed, in some institutions, patients may be emergently evaluated and urgently listed for lung transplantation candidacy during an otherwise lethal acute exacerbation. Very few limited studies have addressed the problem of outcomes after a lung transplant in stable condition as compared with an acute exacerbation condition [25,26,27] and concluded with a recommendation of still going on towards transplant even during the acute worsening of the clinical condition. However, these experiences are limited to single-center retrospective studies with short follow-up, until 1 year of observation after transplant and without long-term mortality analysis. In a recent study by Dotan Y. et al., conducted in 89 patients with IPF, survival analysis was obtained comparing 52 patients transplanted during stable IPF versus during acute exacerbation of IPF. The cohort of patients with AE-IPF had lower survival at one year as well as in the second and third-year post-transplantation compared to patients transplanted during stable IPF. This observation raises important clinical and ethical questions. In fact, it can be argued that not transplanting the sickest patients among acute exacerbators would have been a “death sentence”; however, on the other hand, transplanting the sickest patients leads to less survival benefit. The authors concluded that patients with AE-IPF and very high LAS (lung allocation score) might not experience the survival advantage expected from lung transplantation [28]. Given the shortage of lung donors, further long-term studies will be helpful to deeply ascertain the benefit-risk ratio and refine the decision-making process for each patient.

## 4. Timing for Transplant: the LAS (Lung Allocation System) Issue

As the demand for lung allografts increases, it is crucial to optimize the use of limited resources by selecting recipients with the best prospects of positive long-term outcomes. In the previous decades, lungs were allocated on the base of the starting date on the waiting list. Because of the inequities of this time-based system and the high morbidity and mortality rate, to identify the best candidates for transplant was challenging for clinicians. As a matter of fact, physicians have investigated various measures to further optimize the listing criteria for LTx. Age, weight, FVC% predicted, the amount of oxygen required at rest, and outcomes on hospitalization, intensive care unit (ICU) admission, mechanical ventilation, wedge pressure on catheterization, systolic blood pressure, and psychologic conditions have been shown as predictors of mortality in IPF patients before transplant [29,30,31]. Mechanical support at transplant, history of coronary artery disease at the listing, pCO_2_ at transplant are considered as predictors of post-transplant survival in IPF [30].

The lung allocation system (LAS) is an allocation system based on a complex risk-benefit ratio of expected wait-list urgency to expected post-transplant survival and which essentially reflects disease severity. LAS is calculated using different measures of patient’s health, which estimate survival probability in the waiting list period and the post-transplant period (Table 3). Higher LAS scores greater than 46 predict worse post-transplant survival [32]. The beginning of the ‘LAS era’ resulted in an increased number of IPF patients receiving a lung transplant in the United States and in shorter times spent in the wait-list, as compared to the patients in the waiting list belonging the pre-LAS period [31]. Despite the lower median wait-list time to transplant and the higher transplant rate among IPF patients as compared to other pulmonary diseases, Organ Procurement and Transplantation Network (OPTN) have shown that the pre-transplant mortality rate among adult IPF patients wait-listed for a lung transplant remains higher as compared to other pulmonary diseases [33]. However, there are far fewer available donor organs than patients who would potentially benefit from lung transplantation. Despite the introduction of the LAS, the risk of death remains still high due to the unpredictable course of IPF or the occurring of acute exacerbations. The optimal use of resources is crucial, and patients should be selected for transplantation when they have still the best chances for favorable long-term outcomes.

*Survival benefits*. In the most recent publication from ISHLT, the median survival after lung transplant for patients with interstitial lung diseases (ILD) is 4.7 years. This is significantly less than post-transplant survival for chronic obstructive pulmonary disease (COPD) and cystic fibrosis (CF) patients (and 5.5 and 8.3 years, respectively). The available data from the OPTN and ISHLT demonstrate that approximately 50% of IPF patients are alive at five years post-transplantation. IPF patients have poorer survival as compared to lung transplant patients with other underlying diagnoses. Potential reasons for this survival disparity may include older age at the time of transplant, a relatively higher proportion of single lung transplants, increased prevalence of age-related comorbidities, and higher prevalence of bronchiolitis obliterans in this population [34,35]. Pirfenidone and Nintedanib have been recently established to slow down the functional decline in IPF patients, but only a few reports have investigated if they may interfere with wound-healing after major surgery, especially in the bronchial anastomosis formation after lung transplantation. Up to now, the real-life observational study has shown that previous use of anti-fibrotic therapy does not increase surgical complications or post-operative mortality after LTx [36,37]. It remains to be elucidated if antifibrotic therapy may influence the referral and the subsequent decision on listing for transplant. This issue should be addressed in further observational studies from transplant centers in order to figure out homogeneous criteria and capture the right patients for the surgical program.

## 5. Management Before LTx

Once listed for LTx, it is mandatory to maintain the functional status of the patients as best as possible. To purpose this aim, it is extremely important that all patients participate actively in a supervised pulmonary rehabilitation program [38,39]. We have previously described that factors predictive of IPF post-transplant mortality include history of coronary artery disease at the time of listing, PCO_2_ at the time of transplant, and mechanical ventilation at transplant [30], and recent evidence has added pulmonary hypertension and high or low body mass index to the list [40,41,42,43].

Patients with IPF tend to have worse conditions as compared to patients affected by other causes of end-stage lung disease and waiting for LTx. In fibrotic patients, the time in wait-list is relatively long as compared to the rapid functional decline and the few therapeutic options. Advancements in Human Leukocyte Antibody (HLA) testing and the use of extracorporeal membrane oxygenation (ECMO), as a bridge to lung transplantation in carefully selected patients, have improved recipient selection standards in the last years [31]. ECMO allows to retard deconditioning in patients on a waiting list and to reduce preoperative mortality during the waiting time for the organ [44]. Moreover, in ECMO-treated patients, the aggressive postoperative rehabilitation program improves not only perioperative outcomes but also long-term survival [45]. Unfortunately, ECMO is mostly indicated in young patients (younger than 60 years old) with a high residual rehabilitation potential [23,45]; thus, elderly patients with IPF are now not to be considered.

### The Impact of Comorbidities on Mortality–Implication for Transplant

IPF is associated with various respiratory and non-respiratory comorbidities, which are responsible for the unpredictable natural course of the disease, affecting the quality of life and prognosis of each patient [46] (Table 4).

Almost 90% of the patients with IPF present at least one comorbidity. Lung cancer is the most relevant comorbidity that may affect IPF patients. Indeed, patients with IPF have a five-fold increased risk as compared to the general population to develop lung cancer, with the incidence of lung cancer in patients with IPF ranging from 3% to 22%, and in some cases, exceeding 50% of patients during IPF clinical course [47,50]. IPF and lung cancer share several risk factors and also pathogenic pathways. The concomitant conditions are associated with significantly worse prognosis, which is almost halved [51]. For all these reasons, in transplant candidates with IPF, strict radiological surveillance is recommended to exclude the candidates who develop lung cancer during the wait-list. Nevertheless, even when strict radiological monitoring is made, cancer may not be detectable on CT imaging because of the huge amount of confluent fibrosis visible in the progressive stage of the disease, and which may hide the presence of lung nodules or small area of ground-glass opacities. Notably, several authors have described lung recipients whose explanted lungs harbor unexpected cancer and which may affect survival after transplant [52,53]. Taking into consideration these concomitant conditions, clinicians must be keenly aware of the possibility of unexpected neoplasms and should carefully evaluate new or growing lung nodules in the CT scans of recipients. Moreover, the CT scan can detect emphysema, which is considered to worsen clinical outcome in IPF [22].

Cardiovascular disease, including coronary artery diseases and pulmonary hypertension, are other relevant comorbidities found in patients with IPF and associated with a reduction of survival [46,49]. Even without the existence of specific therapy to treat pulmonary hypertension [2], an accurate evaluation of the mean pulmonary artery pressure and right ventricular function is mandatory to assess the mortality risk before transplant and to plan the correct anesthesiologic management during transplantation. As a matter of fact, a positive correlation between mean pulmonary pressure and survival has been shown, with mortality increasing with higher levels in pulmonary pressure [54]. Similarly, high mean pulmonary pressure is associated with an increased incidence of primary graft dysfunction and early postoperative mortality after transplant [41]. Finally, previous data have shown that patients with IPF have a marked relative increase in the risk of vascular disease [55]. Preoperative screening is essential in patients with IPF and the presence of coronary artery disease amenable to revascularization might not exclude transplantation. However, patients with the multivessel disease or impaired left ventricular function are generally considered ineligible.

## 6. Conclusions

In the new era of accessible medical therapy for IPF, the need for lung transplantation may be delayed, but it will remain the definitive treatment for advanced end-stage disease. Whether or not the newly available medications for IPF will change the course of the disease and affect the number of IPF patients being listed and transplanted remains to be determined. No single criterion is enough to predict prognosis: experiences and outcomes vary among transplant centers and depend also on organ allocation systems. Early referral to a transplant center is strongly recommended in order to discuss with the patient and family the risk/benefit balance of this surgical option.

## Figures and Tables

**Figure 1 medicina-55-00702-f001:**
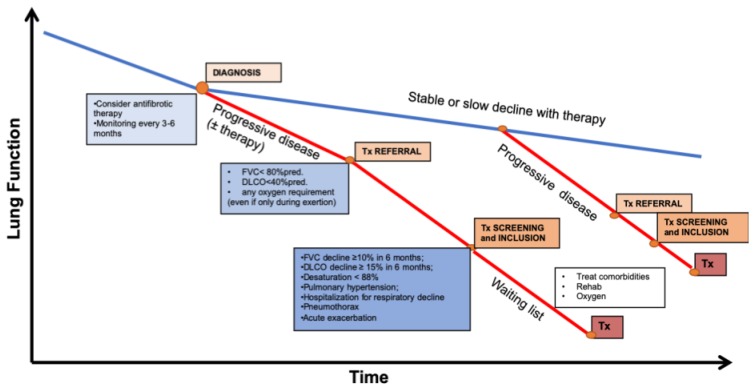
Representation of the steps for the transplant process throughout the natural history of patients with idiopathic pulmonary fibrosis. Tx = transplantation, FVC = forced vital capacity, DLCO = diffusion capacity of the lung for carbon monoxide.

**Table 1 medicina-55-00702-t001:** International Society of Heart and Lung Transplantation criteria for referral and listing of patients with interstitial lung disease (ILD) for lung transplantation.

Criteria for Referral	Criteria for Listing
Histopathologic or radiographic evidence of usual interstitial pneumonitis (UIP) or fibrosing non-specific interstitial pneumonitis (NSIP), regardless of lung function. Abnormal lung function: forced vital capacity (FVC) <80% predicted, or diffusion capacity of the lung for carbon monoxide (DLCO) <40% predicted. Any dyspnea or functional limitation attributable to lung disease. Any oxygen requirement, even if only during exertion. For inflammatory interstitial lung disease (ILD), failure to improve dyspnea, oxygen requirement, and/or lung function after a clinically indicated trial of medical therapy.	Decline in forced vital capacity ≥10%, or diffusion capacity of the lung ≥15% during 6 months of follow-up Desaturation to <88% or distance <250 m on a 6-min walk test, or >50 m decline in 6-min walk distance over 6 months Pulmonary hypertension on right heart catheterization or echocardiography Hospitalization because of respiratory decline, pneumothorax, or acute exacerbation

Reprinted from: A consensus document for the selection of lung transplant candidates: 2014—An update from the Pulmonary Transplantation Council of the International Society for Heart and Lung Transplantation. David Weill, Christian Benden, Paul A., Corris, John H., Dark, R., Duane Davis, Shaf Keshavjee, David J., Lederer, Michael J., Mulligan, G., Alexander Patterson, Lianne G., Singer, Greg I., Snell, Geert M., Verleden, Martin R., Zamora, Allan, R. Glanville. Journal Heart Lung Transplant 2015; 34: 1–15. Copyright (2019), with permission from Elsevier.

**Table 2 medicina-55-00702-t002:** International Society of Heart and Lung Transplantation major and relative contraindications for listing patients for lung transplantation.

Absolute	Relative
Malignancy in the last 2 years Untreatable advanced dysfunction of another major organ system Non-curable chronic extra-pulmonary infection or evidence of active mycobacterium tuberculosis infection. Significant chest wall/spinal deformityClass II or III obesity (body mass index > 35.0 kg/m^2^). Documented non-adherence or inability to follow through with medical therapy or office follow-up, or both Untreatable psychiatric or psychological condition associated with the inability to cooperate or comply with medical therapy Active substance addiction or within the last 6 monthsAbsence of a consistent or reliable social support system	Age older than 65 years. Severely limited functional status with poor rehabilitation potential. Critical or unstable clinical conditionProgressive or severe malnutrition and symptomatic osteoporosis. Colonization with a highly resistant or virulent pathogenic agent. Class I obesity (body mass index 30.0–34.9 kg/m^2^), particularly truncal (central) obesity.

Reprinted from: A consensus document for the selection of lung transplant candidates: 2014—An update from the Pulmonary Transplantation Council of the International Society for Heart and Lung Transplantation. David Weill, Christian Benden, Paul A., Corris, John H., Dark, R., Duane Davis, Shaf Keshavjee, David J., Lederer, Michael J., Mulligan, G., Alexander Patterson, Lianne G., Singer, Greg I., Snell, Geert M., Verleden, Martin R., Zamora, Allan R., Glanville. Journal Heart Lung Transplant 2015; 34: 1–15. Copyright (2019), with permission from Elsevier.

**Table 3 medicina-55-00702-t003:** Clinical and functional factors used to calculate LAS (lung allocation system) [30] for all diseases.

Factors Used to Predict Waiting List Survival	Factors Used to Predict Post-Transplant Survival
FVC % predicted systolic blood pressure O_2_ (L/min) at rest age at transplant body mass index (BMI) New York Heart Association (NYHA) functional status diagnosis 6-min walk distance <150 feet continuous mechanical ventilation diabetes	FVC % predicted Pulmonary Capillary Wedge (PCW) mean pressure ≥20 mmHg continuous mechanical ventilation age at transplant serum creatinine (mg/dL) NYHA functional status diagnosis

**Table 4 medicina-55-00702-t004:** The impact of comorbidities on mortality in patients with IPF.

Disease	Prevalence–Impact on Mortality
Lung cancer	Prevalence 3–22%
	The median survival of 38.7 months [47]
Pulmonary hypertension	Prevalence 3–86%
	Median 1-year survival ranges from 4.8 years (<35 mmHg) to 0.7 years (>50 mmHg). An increase of 10 mmHg was associated with a shortened survival (RR 1.34, *p* < 0.001) [48]
Coronary artery disease	Prevalence 3–68%
	The median survival of 1 year and a half from the time of left catheterization [49]

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
