# Peer review of "Idiopathic Pulmonary Fibrosis and Lung Transplantation: When it is Feasible"

_medicina, 2019, doi:10.3390/medicina55100702_

Round 1
Reviewer 1 Report
The manuscript submitted for consideration by Elisabetta Balestro et al., review the specific challenges and strategies related to lung transplantation in patients with IPF and related comorbidities impacting on post transplant outcomes. Overall, the manuscript is well organized and well written. The topic is of high clinical significance. No more changes seem necessary unless the authors wish to update/add more info based on recently published work.
Author Response
"Please see the attachment."

Reviewer 2 Report
It’s great idea to have early referral and close 24 longitudinal follow-up for potential lung transplant candidates. However the shortage of organ donation and complications from immunosuppressant still rely on the evaluation from transplant surgeon and criteria from lung transplant. Similar to kidney transplant, the average of waiting time for patient with blood type O is 8-10 years. Fortunately, peritoneal dialysis and hemodialysis are available to replace renal function and gain extra time for transplantation. It’d be more useful to have new therapeutic target treatment to prevent the deterioration of lung function before they are reaching the appropriate time for transplant.
need to correct the size of text body, they are in different size. it would be great if you can pay more attention before the submission and correct some of overt mistake in the text and table it would be nice to compare the criteria from different transplant center, the mortality and comorbidity.Author Response
"Please see the attachment."

Reviewer 3 Report
General comment
The review by Balestro et al is of sure interest for the readers, is comprehensive and up to date, reflecting the complexity of lung transplantation in terms of adequate selection of patients and timing.
I have a couple of minor comments on the current version of the manuscript:
1) Although the authors mention the problem of age as a limiting factor in the choice of candidates (page 4, line 130), they do not stress enough that this is indeed one of the most important exclusion criteria. At international level there are patients advocacy initiatives to fight exclusion criteria based on absolute age thresholds. The authors could discuss this point more extensively and report on age-related complications and outcomes.
2) The impact of comorbidities on mortality – implication for transplant. Emphysema combined with fibrosis has a huge impact on survival in IPF. Are there data on the impact of emphysema (or CPFE) on transplant outcomes?
3) The authors correctly report on the impact of acute exacerbation on lung transplant outcome and survival, which is controversial.They state "In fact, not transplanting the sickest patients would cause almost 100% mortality in the acute exacerbation group". Do they have specific data to quote? Moreover, Are there data or studies on ECMO as a bridging strategy to lung transplantation in patients with acute deterioration or exacerbation?
4) English needs editing. The following sentences are not clear and should be reworded (see my comments below):
-Abstract: .....therefore candidates surveillance is crucial to avoid the worst organ allocation even considering the difficulty in detecting small nodules in end stage lungs (beside editing, is this sentence really necessary i the abstract?).
-It remains to be elucidated if antifibrotic therapy with heterogeneity (??) in responsiveness may influence and in which way (EXTENT??) the referral and subsequent decision on listing for transplant. This issue should be addressed in further observational studies from (BY???) transplant center(s??) in in order to figure out homogeneous criteria and capture the right patients for the surgical program (what do you mean? which program? procedure?).
Author Response
"Please see the attachment."

Round 2
Reviewer 2 Report
The shortage of organ donation and complications from immunosuppressant still rely on the evaluation from transplant surgeon and criteria from lung transplant. The manuscript is well written with updated information after revision. Understanding the difficult for multiple center data collection, I have no further suggestions at this moment.